# Long non-coding RNA PSMB8-AS1 as a potential biomarker for postoperative recurrence in patients with Fuhrman grades 1–3 clear cell renal cell carcinoma

Chieko Baba[1], Hiroshi Hirata[ORCID][1]*, Koichiro Hiyoshi[1], Takanori Tokunaga[1], Nakanori Fujii[1], Kosuke Shimizu[1], Keita Kobayashi[1], Takahide Hayano[ORCID][2,4], Yoshiyuki Asai[2,3,4], Koji Shiraishi[1]

1 Department of Urology, Graduate School of Medicine, Yamaguchi University, Ube, Yamaguchi, Japan, 2 Department of Systems Bioinformatics, Graduate School of Medicine, Yamaguchi University, Yamaguchi, Japan, 3 AI Systems Medicine Research and Training Center, Graduate School of Medicine and University Hospital, Yamaguchi University, Yamaguchi, Japan, 4 The Division of Systems Medicine and Informatics, Research Institute for Cell Design, Medical Science, Yamaguchi University, Yamaguchi, Japan

* hiro1333@yamaguchi-u.ac.jp

## Abstract

Long non-coding RNAs (lncRNAs) are important regulators of oncogenesis. In this study, we investigated the tumor-promoting role and prognostic value of the lncRNA PSMB8-AS1 in renal cell carcinoma (RCC). Using lncRNA microarray analysis, we identified PSMB8-AS1 as a candidate gene associated with disease progression and immune checkpoint inhibitor resistance. We examined PSMB8-AS1 expression in tumors and adjacent normal renal tissues from 192 patients with RCC. *In vitro* functional assays were performed to assess their role in cell proliferation and invasion. We also conducted bioinformatics and luciferase reporter assays to clarify the interaction between miR-204-5p/miR-211 and the transcription factor TFAP2A. PSMB8-AS1 was significantly overexpressed in clear cell RCC (ccRCC) tissues and correlated with poor prognosis. Knockdown experiments revealed that PSMB8-AS1 promoted proliferation and invasion. Mechanistically, PSMB8-AS1 functions as a competing endogenous RNA, sponging miR-204-5p/miR-211 and enhancing TFAP2A expression. These findings suggest that PSMB8-AS1 represents a promising biomarker for postoperative ccRCC recurrence.

## Introduction

Renal cell carcinoma (RCC) accounts for approximately 2% of all malignancies worldwide, and is the third most prevalent urological cancer after prostate and bladder cancers [1]. In the United States alone, RCC accounted for an estimated 81,800 new cases and 14,890 deaths in 2023 [2]. The 5-year survival rate markedly differs

**Data availability statement:** All relevant data for this study are publicly available from the NCBI Gene Expression Omnibus (GEO) repository (https://www.ncbi.nlm.nih.gov/geo/query/acc.cgi?acc=GSE319768).

**Funding:** The author(s) received no specific funding for this work.

**Competing interests:** The authors have declared that no competing interests exist.

by disease stage, ranging from 81% for stage I to 8% for stage IV [3]. The prognosis remains especially poor in patients with metastases or recurrent tumors.

Advances in systemic therapies have improved outcomes [4]. While cytokine therapies (e.g., IFN-α, IL-2) are effective for select patients, targeted therapies, including tyrosine kinase inhibitors (TKIs) and mTOR inhibitors, have broadened treatment strategies. More recently, immune checkpoint inhibitors (ICIs) that target the PD-1/PD-L1 pathway have shown clinical benefits in restoring antitumor immunity [5].

LncRNAs, defined as transcripts longer than 200 nucleotides without protein-coding capacity, are increasingly being implicated in cancer development and progression [6]. Several genes, including *NEAT1*, *HOTAIR*, *TUG1*, *MALAT1*, and *PVT1*, have been associated with RCC prognosis [7].

In our initial screening, using lncRNA microarray data from collecting duct carcinoma (CDC), a highly aggressive RCC subtype, we found that PSMB8-AS1 was significantly upregulated in tumors that were unresponsive to ICI therapy. CDC and ccRCC arise from distinct nephron segments; however, both may exhibit overlapping molecular features in ICI-resistant settings. This prompted us to evaluate the functional and prognostic relevance of PSMB8-AS1 in patients with ccRCC.

Here, we analyzed PSMB8-AS1 expression in a cohort of 192 patients with ccRCC and investigated its clinical significance and biological function by focusing on its interaction with miRNAs and downstream targets such as TFAP2A.

## Materials and methods

### Microarray lncRNA expression profiling from CDC samples

To identify lncRNAs that are highly expressed in RCC tissues and are potentially associated with disease development and recurrence, we performed microarray analysis using formalin-fixed, paraffin-embedded (FFPE) samples from patients with CDC, a highly aggressive RCC subtype. RNA was extracted from the cancerous and adjacent normal kidney tissues of two patients: one with a durable response to ICI therapy for more than 6 months, and the other with early treatment failure. We focused on lncRNAs that were upregulated in non-responder tumor tissues.

FFPE sections were stained with hematoxylin and eosin to identify tumor and normal regions, which were then microdissected. Total RNA was extracted using a Maxwell RSC RNA Tissue Kit (Promega, Madison, WI, USA). The RNA quantity was assessed using a NanoDrop One spectrophotometer (Thermo Fisher Scientific, Waltham, MA, USA), and RNA integrity was confirmed using DV200 values. Using 100 ng of total RNA per sample, lncRNA expression profiling was performed using the Arraystar Human LncRNA Expression Array V5.0 (ArrayStar Inc., Rockville, MD, USA), which covers 39,317 annotated lncRNAs. RNA labeling and hybridization were performed according to the manufacturer's instructions. Slides were scanned using an Agilent G2505C scanner, and signal intensities were extracted using Agilent Feature Extraction software. Quantile normalization was applied and differential expression was determined using t-tests. lncRNAs with fold change> 2.0, and p-value <0.05 were considered significant. Hierarchical clustering and principal component analysis (PCA) were used for visualization.

## Clinical samples

This study included 192 patients with pathologically confirmed ccRCC who underwent partial or radical nephrectomy at Yamaguchi University Hospital between 2005 and 2023.

Access to clinical data and biological samples for research purposes was conducted from 24/10/2018–31/03/2023. The patient characteristics are summarized in S1 Table. All of the participants provided written informed consent. The study protocol was approved by the Institutional Review Board of Yamaguchi University (Approval No. H30-089-3), and complied with the Declaration of Helsinki.

## Cell culture and chemicals

Two human RCC cell lines, 769-P [CRL-1933] and ACHN [CRL-1611]), and a normal renal proximal tubular epithelial cell line (RPTEC [PCS-400–010]) were obtained from the American Type Culture Collection (ATCC). RCC cell lines were maintained in Gibco RPMI 1640 (Thermo Fisher Scientific) supplemented with 10% fetal bovine serum (FBS), whereas RPTEC cells were cultured in Renal Epithelial Cell Growth Medium BulletKit (Lonza, Basel, Switzerland). All cultures were maintained at 37°C in a 5% $CO_2$ humidified incubator.

## RNA extraction and quantitative RT-PCR

Total RNA was extracted from FFPE tumors and matched to adjacent normal tissues using the miRNeasy FFPE Kit (Qiagen, Hilden, Germany). RNA was extracted from the cell lines using the miRNeasy Mini Kit (Qiagen). Reverse transcription was performed using a PrimeScript RT Reagent Kit (Perfect Real-Time; Takara Bio, Shiga, Japan). Quantitative PCR was conducted using the TaqMan Fast Advanced Master Mix and specific TaqMan probes (Applied Biosystems, Carlsbad, CA, USA) on a QuantStudio 3 instrument (Thermo Fisher Scientific). ACTB and RNU48 served as endogenous controls. Relative expression was calculated using the $2^{-\Delta Ct}$ method. The TaqMan MicroRNA Reverse Transcription Kit (Thermo Fisher Scientific) was used for miRNA analysis.

## siRNA-mediated knockdown

ACHN and 769-P cells were seeded in 6-well plates and transfected with PSMB8-AS1-specific siRNA (si-PSMB8-AS1) or negative control siRNA (si-NC) using Lipofectamine RNAiMAX (Invitrogen; Thermo Fisher Scientific). The siRNAs were purchased from Silencer Select (Ambion, CA, USA).

## TFAP2A overexpression (rescue) after PSMB8 knockdown

To evaluate whether TFAP2A overexpression could rescue the phenotypic changes induced by PSMB8 knockdown, TFAP2A expression was restored using a TFAP2A expression vector. Briefly, cells were first transfected with PSMB8-specific siRNA using RNAiMAX transfection reagent according to the manufacturer's instructions. After 6 hours, cells were transfected with a human TFAP2A expression plasmid (pRP-CAG-hTFAP2A; VectorBuilder) using X-tremeGENE HP DNA Transfection Reagent. An empty vector was used as a negative control. TFAP2A overexpression was confirmed by RT-qPCR prior to subsequent functional assays, including cell proliferation and invasion assays.

## Cell proliferation experiments

The transfected cells were plated in 96-well plates. Cell Titer 96 Aqueous One Solution Cell Proliferation Assay (Promega) was added to the cells after 24 h, 48 h, and 72 h. After incubation for a further 2 h, the absorbance of each well was measured using a microplate reader (iMark; Bio-Rad, Hercules, CA, USA) at 490 nm.

## Cell invasion assay

The invasion capacity was assessed using a CytoSelect 24-well cell invasion assay kit (Cell Biolabs, San Diego, CA, USA). Cells ($1.0 \times 10^6$/mL) in serum-free medium were seeded into the upper chamber of a Matrigel-coated insert. Medium containing 10% FBS was added to the lower chamber. After incubation, the cells on the lower surface were stained and quantified by measuring the absorbance at 560 nm.

## Luciferase reporter assay

To validate direct interaction between miRNAs and TFAP2A, the 3′UTR of TFAP2A (wild-type and mutant constructs) was cloned into the pmirGLO Dual-Luciferase miRNA Target Expression Vector (Promega). Plasmids for 3′UTR luciferase assays were made as described previously [8]. The primer sequences included PmeI and XbaI restriction sites. ACHN and 769-P cells were co-transfected with these constructs and synthetic miR-204/211 mimics using Lipofectamine 2000. Luciferase activity was measured at 48 h post-transfection using the Dual-Luciferase Reporter Assay System (Promega). The results are presented as the mean ± SD from triplicate experiments.

The primer sequences used were as follows: TFAP2A forward primer, 5′ AAACTA GCGGCCGCTAGTtcTTGC CCCCTTTGACCTTCcT 3′; TFAP2A reverse primer, 5′ CTAGAgGAAGGTCAAAGGGGGCAAgaACTA primer, mutant TFAP2A forward primer, 5′AAACTAGCGGCCGCTAGTtcCTTGGCGCTCTTCGGCCGcT 3′; and mutant TFAP2A reverse primer, 5′ CTAGAgCGGCCGAAGAGCGCCAAGgaACTA GCGGCCGCTAGTTT 3′.

## Bioinformatics analysis

To identify miRNAs that bind PSMB8-AS1, we used the miRanda tool with the ensemble transcript (ENST05220043563.1; 3310 bp) as input. Mature miRNA sequences for Homo sapiens were downloaded from miRBase. Binding predictions for miRNAs and their mRNA targets (e.g., TFAP2A) were performed using TargetScan and miRDB. A pairing score ≥160 and energy ≤−20 kcal/mol were used as cutoff thresholds.

## Statistical analysis

Cutoff values for high vs. low PSMB8-AS1 expression were determined by receiver operating characteristic (ROC) analysis based on progression-free survival (PFS), with thresholds selected using the Youden index. Categorical variables were compared using chi-square tests, while continuous variables were analyzed using Student's t-test or the Mann–Whitney U test, as appropriate. For analyses of paired tumor and adjacent normal tissue samples, the Wilcoxon signed-rank test was used.

Survival was estimated using Kaplan–Meier analysis and compared using the log-rank test. Multivariate Cox proportional hazards regression was performed to assess the independent predictors of PFS. All analyses were conducted using JMP Pro 16 software (SAS Institute, Cary, NC, USA). Statistical significance was set at $P < 0.05$.

## Results

### Identification of PSMB8-AS1 as a candidate lncRNA in RCC

Microarray analysis identified 1,342 lncRNAs that were upregulated in ICI-nonresponsive CDC samples (Fig 1A). Among these candidates, PSMB8-AS1 was selected for further analysis based on its consistent upregulation across both ICI responder and non-responder tumors, relatively high fold-change compared with other candidates, and limited prior functional characterization in RCC.

### PSMB8-AS1 expression in clinical samples and its prognostic relevance

Expression levels of PSMB8-AS1 were compared between paired tumor (T) and adjacent normal (N) tissues obtained from the same patients (Fig 1B; n = 173). Each dot represents an individual sample, and paired samples are connected by

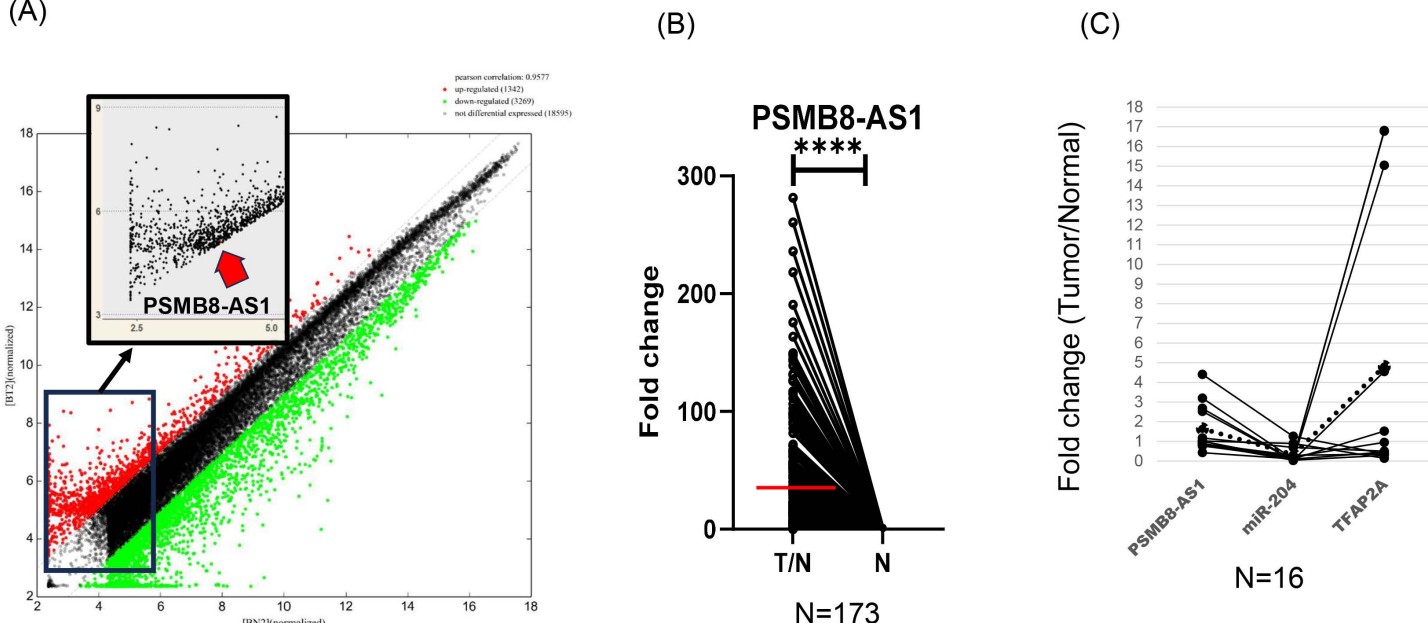

**Fig 1. lncRNA array data and expression of PSMB8-AS1 in clinical samples.** (A) Scatter plot shows that 1342 lncRNAs were upregulated in ICI-nonresponsive collecting duct carcinoma cases. **(B)** Paired expression analysis of PSMB8-AS1 in tumor and normal tissues (n=173). Expression levels of PSMB8-AS1 were compared between paired tumor (T) and adjacent normal (N) tissues obtained from the same patients. Each dot represents an individual sample, and paired samples are connected by lines. Expression levels are presented as fold change relative to normal tissue for visualization. **(C)** Quantitative RT-PCR analysis of PSMB8-AS1, miR-204, and TFAP2A expression in paired tumor (T) and adjacent normal renal (N) tissues from ccRCC patients (n = 16). Differences between paired tumor and adjacent normal tissues were evaluated using the Wilcoxon signed-rank test. NS, not significant; *P < 0.05; **P < 0.01; ***P < 0.001; ****P < 0.0001.

lines. Expression levels are presented as fold change relative to normal tissue for visualization. Statistical comparisons were performed using the Wilcoxon signed-rank test based on ΔCt values.

Quantitative RT-PCR analysis of PSMB8-AS1, miR-204, and TFAP2A expression in paired tumor (T) and adjacent normal renal (N) tissues from ccRCC patients (n = 16; Fig 1C). Each dot represents an individual patient sample, and paired samples are connected by lines. Differences between paired tumor and adjacent normal tissues were evaluated using the Wilcoxon signed-rank test. NS, not significant; *P < 0.05; **P < 0.01; ***P < 0.001; **** indicates P < 0.0001.

Kaplan–Meier analysis revealed that patients with high PSMB8-AS1 expression had significantly shorter PFS. Stratified subgroup analyses showed that elevated PSMB8-AS1 expression was associated with worse PFS in patients with pT3/pT4 disease, nodal involvement (N1/N2), distant metastases (M1), Fuhrman grade 4 disease, and high lncRNA expression levels (Fig 2).ROC analysis identified 17.4 as the optimal cutoff value for PSMB8-AS1 expression to predict PFS and overall survival (OS) across the full cohort (sensitivity, 80%; specificity, 61.08%). In early-stage patients (T1–T3, N0, M0, Fuhrman grades 1–3), a higher cut-off value of 48.55 was found to be more predictive (sensitivity, 63.16%; specificity, 81.71%). For OS, significant differences were observed in pathological stage (pT3+4), nodal and metastatic status, and Fuhrman grade, but not in PSMB8-AS1 expression levels (Fig 3).

## Prognostic significance in early-stage RCC

In the early-stage subgroup (T1–T3, N0, M0, Fuhrman grade 1–3), patients with high PSMB8-AS1 expression had a significantly shorter PFS (log-rank P = 0.001). ROC analysis using a cutoff value of 48.55 confirmed the predictive utility of PSMB8-AS1 in this subgroup (Fig 4).

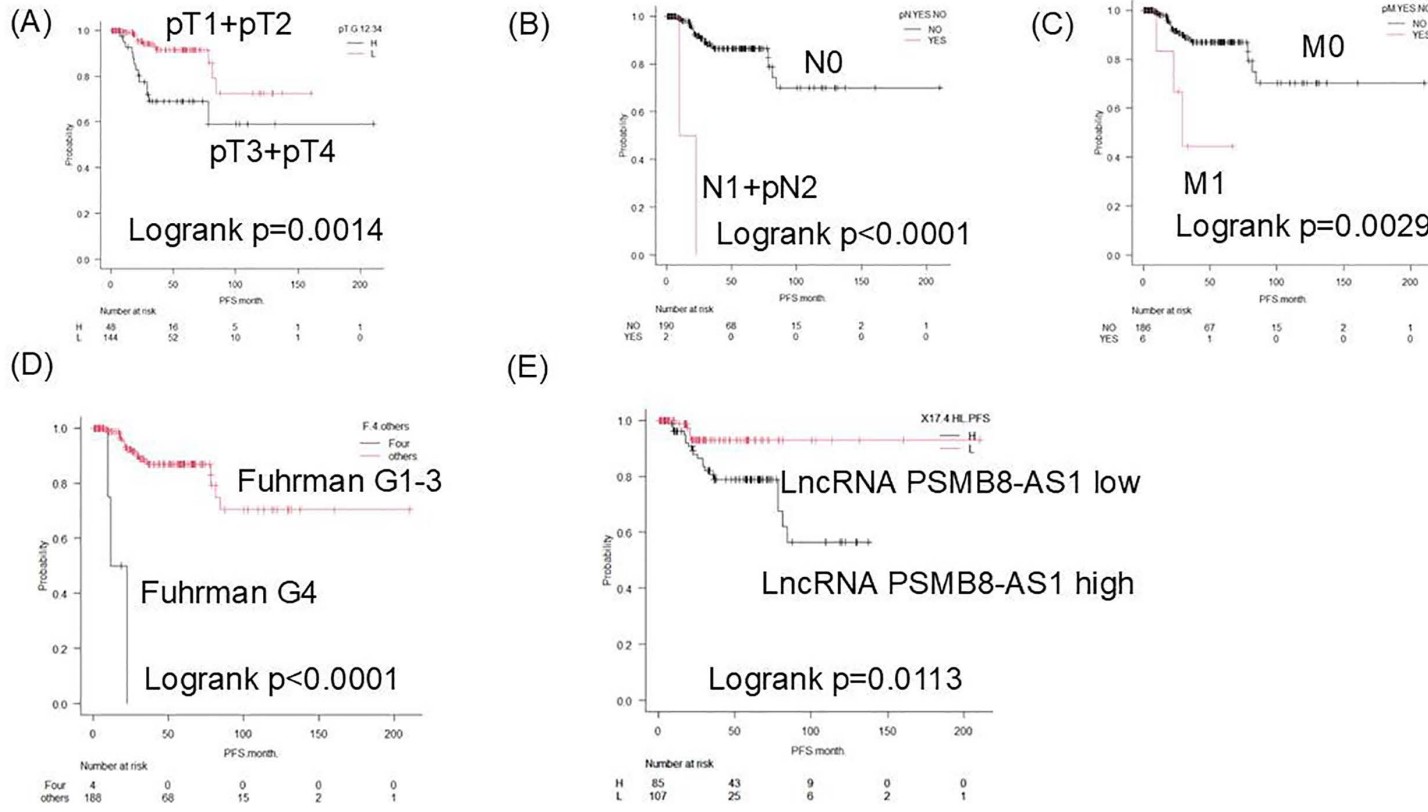

**Fig 2. Kaplan–Meier analysis using several markers for recurrence prediction. pT stage was defined according to the pathological tumor stage (pT1–pT4). pN indicates pathological lymph node status (pN0–pN2), and pM indicates pathological distant metastasis (pM0–pM1).** H and L indicate high and low PSMB8-AS1 expression groups, respectively, as determined by ROC analysis. **(A)** pT3＋pT4 vs. pT1＋pT2. **(B)** pN1＋PN2 vs. pN0. **(C)** pM1 vs. pM0. **(D)** Fuhrman grade 4 vs. grades 1＋2＋3. **(E)** lncRNA PSMB8-AS1 high (H) vs. low (L).

## Multivariate analysis

Cox proportional hazards modeling identified high PSMB8-AS1 expression as an independent predictor of PFS in both the entire cohort (hazard ratio [HR]: 4.28, P＝0.00098) (see S2 Table A) and the early-stage subgroup (HR: 3.43, P＝0.0115) (see S2 Table B). No clinicopathological factors, including PSMB8-AS1 expression, were independently predictive of OS (see S3 Table).

## Functional role of PSMB8-AS1 in RCC cell lines

PSMB8-AS1 was upregulated in the RCC cell lines 769-P and ACHN compared to RPTEC (Fig 5A). siRNA-mediated knockdown of PSMB8-AS1 significantly reduced cell proliferation and invasion in both cell lines (Fig 5B–5D).

## PSMB8-AS1 modulates the miR-204/211–TFAP2A axis

Bioinformatics predictions indicated that PSMB8-AS1 may function as a competing endogenous RNA (ceRNA) for miR-204 and miR-211. Both miRNAs were significantly downregulated in RCC cell lines compared to normal cells (Fig 6A). PSMB8-AS1 knockdown restored the miR-204/211 levels (Fig 6B). TFAP2A, a validated target of miR-204/211, was over-expressed in RCC cells (Fig 6C) and its expression was reduced following PSMB8-AS1 knockdown (Fig 6D).

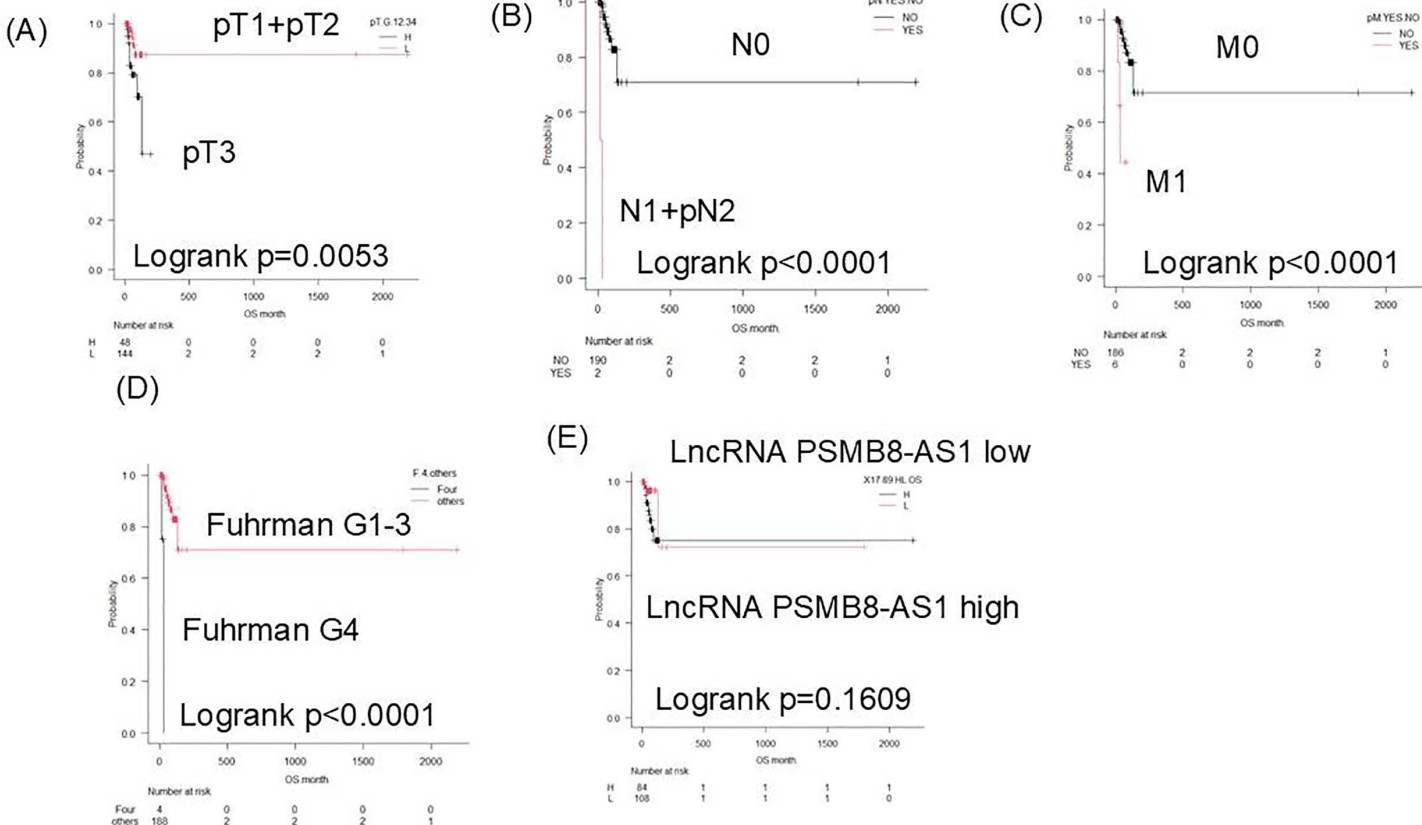

**Fig 3. Kaplan–Meier analysis using several markers for overall survival.** pT stage was defined according to the pathological tumor stage (pT1–pT4). pN indicates pathological lymph node status (pN0–pN2), and pM indicates pathological distant metastasis (pM0–pM1). H and L indicate high and low PSMB8-AS1 expression groups, respectively, as determined by ROC analysis. **(A)** pT3 + pT4 vs. pT1 + pT2. **(B)** pN1 + PN2 vs. pN0. **(C)** pM1 vs. pM0. **(D)** Fuhrman grade 4 vs. grades 1 + 2 + 3. **(E)** lncRNA PSMB8-AS1 high (H) vs. low (L).

Luciferase reporter assays confirmed direct binding of miR-204/211 to the TFAP2A 3′UTR. Co-transfection of RCC cells with miR-204/211 mimics and wild-type TFAP2A 3′UTR vectors resulted in decreased luciferase activity, whereas mutant vectors abolished this suppression (Fig 6E). These results support the idea that the PSMB8-AS1–miR-204/211–TFAP2A regulatory axis contributes to RCC progression.

### Functional rescue of RCC cell invasion by TFAP2A following PSMB8-AS1 knockdown

To determine whether the reduced invasive capacity induced by PSMB8-AS1 knockdown was mediated by TFAP2A, rescue experiments were performed. Forced overexpression of TFAP2A significantly restored the invasive ability of RCC cells suppressed by siPSMB8-AS1 transfection. In addition, TFAP2A overexpression (Fig 7A) partially reversed the siPSMB8-AS1–induced suppression of cell proliferation (Fig 7B) and cell invasion (Fig 7C). These findings indicate that TFAP2A functions as a critical downstream effector of PSMB8-AS1 in regulating both invasion and proliferation of RCC cells.

### Discussion

This study is the first to identify PSMB8-AS1 as a clinically relevant lncRNA in ccRCC, demonstrating its prognostic significance and oncogenic function. Initially discovered through microarray analysis of ICI-resistant CDC, PSMB8-AS1 is

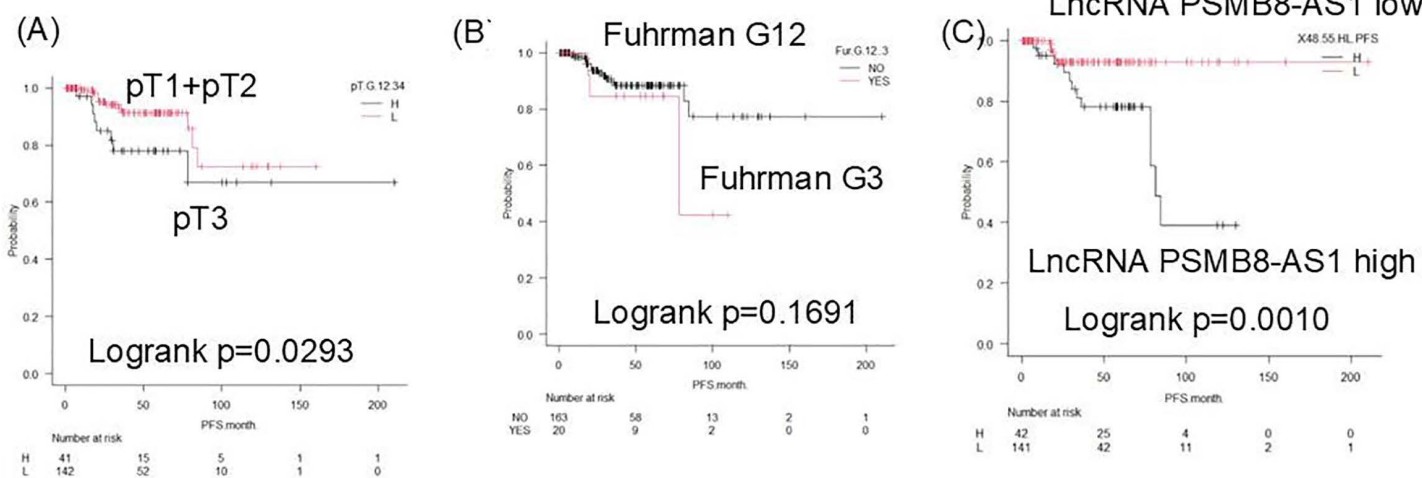

**Fig 4. Kaplan–Meier analysis using several markers for recurrence prediction in patients with pT123N0M0, Fuhrman grade 123. pT stage was defined according to the pathological tumor stage (pT1–pT4). pN indicates pathological lymph node status (pN0–pN2), and pM indicates pathological distant metastasis (pM0–pM1).** H and L indicate high and low PSMB8-AS1 expression groups, respectively, as determined by ROC analysis. **(A)** pT3 vs. pT1 + pT2. **(B)** Fuhrman G3 vs. G1 + G2. **(C)** lncRNA PSMB8-AS1 high (H) vs. low (L).

upregulated in aggressive renal tumors. This has prompted further investigation into ccRCC, which is the most prevalent subtype of RCC.

We showed that PSMB8-AS1 expression was significantly higher in ccRCC tumors than in adjacent normal tissues, and was independently associated with poor PFS. Notably, this correlation was true in both the overall cohort and in early-stage (T1–T3, N0, M0, Fuhrman grades 1–3) patients, suggesting its potential utility as a biomarker for postoperative recurrence, even among individuals with favorable clinical profiles.

Although ROC and Cox regression analyses demonstrated the prognostic relevance of PSMB8-AS1, the clinical applicability of this biomarker should be considered exploratory. Cutoff values were defined within a single cohort and require further validation in independent datasets. Future studies incorporating external validation, integrated multivariable prediction models, and non-invasive detection approaches will be essential to establish the clinical utility of PSMB8-AS1.

The lack of a significant association with OS may reflect several factors, including relatively short follow-up, differences in salvage treatment strategies after recurrence, and the confounding effects of non-cancer-related mortality. Further longitudinal studies are necessary to evaluate the predictive value of PSMB8-AS1 for OS.

Functional assays revealed that PSMB8-AS1 knockdown significantly inhibited cell proliferation and invasion, underscoring its role in tumor aggressiveness. Mechanistically, a regulatory axis involving miR-204/211 and TFAP2A was identified. PSMB8-AS1 acts as a ceRNA, sequestering miR-204/211, which in turn leads to the upregulation of the transcription factor TFAP2A, a known oncogenic factor in RCC [9] and other cancers [10,11]. Per previous reports on miR-204 and TFAP2A as targets of PSMB8-AS1, miR-204 exerts tumor-suppressive effects in renal cancer through multiple molecular pathways [12]. Interestingly, a comprehensive analysis of TFAP2A conducted across 33 cancer types using large-scale databases, such as TCGA, GTEx, TIMER2, and HPA, along with experimental data, revealed that TFAP2A was highly expressed at both the mRNA and protein levels in most cancer types, consistent with a previous study [13].

Importantly, functional rescue experiments demonstrated that TFAP2A overexpression reversed the invasion-suppressive effects of PSMB8-AS1 knockdown, providing direct evidence that the pro-invasive function of PSMB8-AS1 is mediated, at least in part, through the miR-204/211–TFAP2A axis. Although direct experimental validation of PD-L1

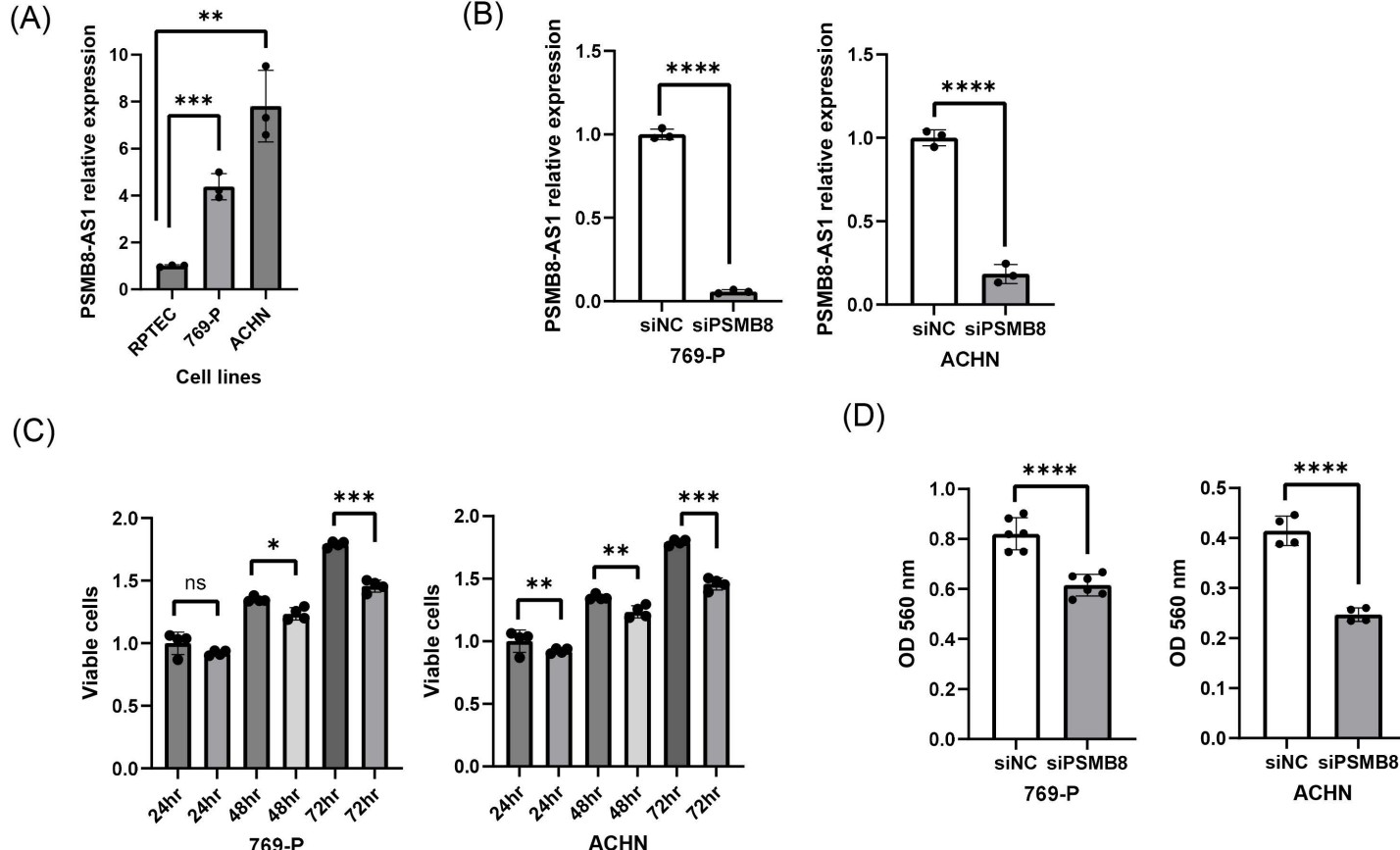

**Fig 5. Effect of lncRNA PSMB8-AS1 knockdown on renal cancer cell function (ACHN and 769-P). (A)** Expression of PSMB8-AS1 was significantly increased in two renal cancer cell lines (ACHN and 769-P) compared to RPTEC. Two renal cancer cell lines were transiently transfected with either si-PSMB8-AS1 or control (si-NC) for further experiments. **(B)** Validation of PSMB8-AS1 knockdown in RCC cell lines. **(C)** Cell viability was assessed using MTS assay. Downregulation of PSMB8-AS1 significantly suppressed cell proliferation in both cell lines 48 h post-transfection. **(D)** Invasion assay. Knockdown of PSMB8-AS1 significantly decreased invasion ability of both cancer cell lines compared with negative control. Data are presented as individual data points with bars indicating the mean ± standard deviation (SD). n = 3 per group. Statistical significance was assessed using one-way ANOVA with Tukey's post hoc test. *$P < 0.05$; **$P < 0.01$; ***$P < 0.001$; ****$P < 0.0001$.

regulation was not performed in this study, previous reports have demonstrated that TFAP2A can transcriptionally activate PD-L1 expression in multiple cancer types. Therefore, the involvement of PD-L1 in the PSMB8-AS1–miR-204/211–TFAP2A axis should be considered a hypothesis that warrants further investigation in future studies.

Although our findings suggest a potential association between PSMB8-AS1 and immune checkpoint resistance, no immune-related assays were performed in this study. Therefore, any conclusions regarding tumor–immune interactions should be interpreted with caution. Future investigations incorporating immune infiltration profiling and functional immune assays will be essential to clarify the role of PSMB8-AS1 in the tumor immune microenvironment.

Although our results demonstrate that PSMB8-AS1 promotes malignant phenotypes in RCC cells, the effects of PSMB8-AS1 suppression in normal renal epithelial cells were not evaluated in this study. Future studies examining potential cytotoxic effects in normal cells will be necessary to further assess the specificity and safety of targeting PSMB8-AS1. Notably, PSMB8-AS1 expression was low in normal renal tissues and RPTEC cells, suggesting a potential therapeutic window; however, this hypothesis requires direct functional validation.

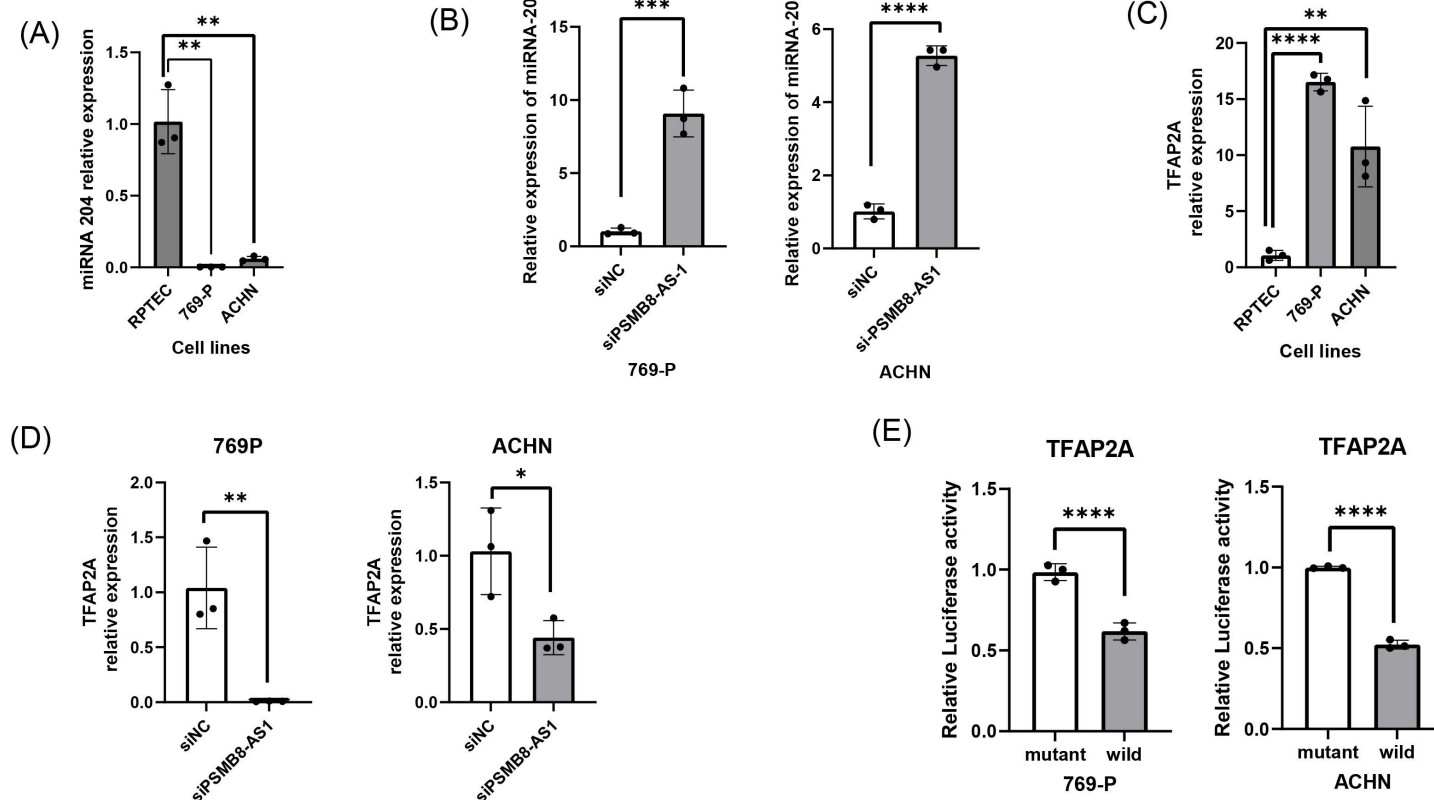

**Fig 6. Effect of lncRNA PSMB8-AS1 knockdown on miR-204 and TFAP2A expression in the RCC cell lines, ACHN and 769-P. (A)** miR-204 expression was significantly lower in the two cancer cell lines compared to the normal kidney cell line. **(B)** Knockdown of PSMB8-AS1 significantly increased miR-204 expression in the two RCC cell lines (ACHN and 769-P). **(C)** TFAP2A expression was significantly higher in the two cancer cell lines compared to the normal kidney cell line. **(D)** Luciferase assay. MiR-204 and 3′UTR vectors with wild type or mutant sequence were co-transfected into two RCC cell lines (ACHN and 769-P). Cell lysates were analyzed for relative luciferase activity 48 h after transfection. The levels of luciferase activity were compared to those of cells transfected with 3′UTR vector with a mutant type sequence. **(E)** PSMB8-AS1 knockdown significantly decreased TFAP2A expression in two RCC cell lines (ACHN and 769-P). Data are presented as individual data points with bars indicating the mean ± standard deviation (SD). n = 3 per group. Statistical significance was assessed using one-way ANOVA with Tukey's post hoc test. *P < 0.05; **P < 0.01; ***P < 0.001; ****P < 0.0001.

In renal cancer, TFAP2A has been suggested to be involved in immune suppression and promotion of PD-L1 expression. Experimental validation using a dual-luciferase assay demonstrated that TFAP2A bound to the PD-L1 promoter and enhanced its transcriptional activity.

PSMB8-AS1, identified as the target gene in this study, was highly expressed in the tumor tissue of a patient with CDC who showed no response to ICIs. If TFAP2A is indeed involved in the immune response, targeting PSMB8-AS1 may ultimately lead to the upregulation of tumor-suppressive miRNA-204, resulting in the downregulation of TFAP2A expression. This, in turn, could reduce the transcriptional activity of PD-L1 and potentially shift the tumor from being resistant to immune checkpoint therapy to becoming responsive.

Our findings align with prior studies reporting oncogenic functions of PSMB8-AS1 in various malignancies, including glioma and pancreatic, colorectal, lung, and bladder cancers [14–16]. The consistency of its tumor-promoting effects across multiple cancer types highlights the broad oncological significance of PSMB8-AS1 as an lncRNA [17, 18].

Despite these promising results, this study has some limitations. First, the retrospective design and single-center cohort may have introduced a selection bias. Second, in vivo validation of the proposed regulatory mechanisms is lacking. This

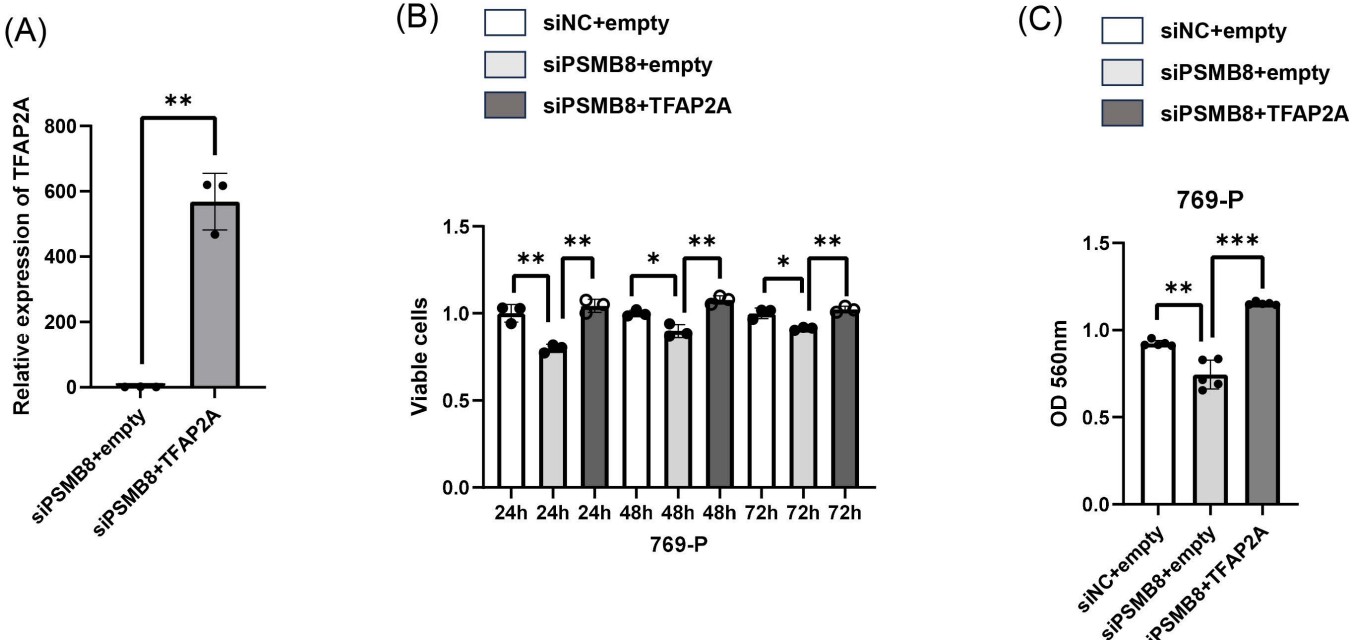

**Fig 7. TFAP2A overexpression rescues the invasive ability suppressed by PSMB8-AS1 knockdown. (A)** TFAP2A overexpression validation (qRT-PCR) **(B)** MTS assay (si-NC+empty/si-NC+TFAP2A/si-PSMB8-AS1+TFAP2A) **(C)** cell invasion assay (si-NC+empty/si-NC+TFAP2A/si-PSMB8-AS1+TFAP2A).

study has several technical limitations. RNA was extracted from FFPE specimens, which may compromise RNA integrity. To minimize this issue, RT-qPCR assays were designed with short amplicons suitable for FFPE-derived RNA. Nevertheless, the absence of detailed RNA quality metrics, such as DV200 values, should be acknowledged. Future studies using fresh-frozen tissues or incorporating additional RNA quality assessments will be important to further validate the robustness of these findings.

Future prospective studies, including animal models, are warranted to establish the clinical and therapeutic relevance of PSMB8-AS1 in RCC. Future transcriptome-wide or proteomic analyses after PSMB8-AS1 knockdown will be necessary to identify additional downstream pathways beyond the miR-204/211–TFAP2A axis.

Another limitation of this study is that functional assays were performed in only two RCC cell lines (769-P and ACHN), which may not fully capture the biological heterogeneity of renal cell carcinoma. Validation in additional experimental models, such as patient-derived cells, organoids, or in vivo xenograft and orthotopic models, will be necessary to determine whether PSMB8-AS1 suppression can consistently inhibit tumor growth and metastasis.

In conclusion, PSMB8-AS1 represents a novel biomarker for early ccRCC recurrence and a potential molecular target for therapeutic interventions. Its involvement in the miR-204/211–TFAP2A axis provides insights into RCC pathogenesis and may inform future strategies for biomarker-driven patient stratification.

## Supporting information

**S1 Table. Patient characteristics.**
(DOCX)

**S2 Table. Univariate and Multivariate analysis to predict recurrence-free survival.**
(DOCX)

**S3 Table. Univariate and Multivariate analysis to predict overall survival.**
(DOCX)

## Acknowledgments

We would like to thank Editage (www.editage.jp) for providing excellent English language editing assistance.

## Author contributions

**Conceptualization:** Chieko Baba, HIROSHI HIRATA, Koji Shiraishi.

**Data curation:** Koichiro Hiyoshi, Takanori Tokunaga, Nakamori Fujii, Takahide Hayano, Yoshiyuki Asai.

**Formal analysis:** Chieko Baba, HIROSHI HIRATA, Nakamori Fujii, Takahide Hayano.

**Investigation:** Chieko Baba, HIROSHI HIRATA, Takanori Tokunaga, Takahide Hayano.

**Methodology:** Chieko Baba, HIROSHI HIRATA, Kosuke Shimizu.

**Project administration:** HIROSHI HIRATA.

**Software:** HIROSHI HIRATA.

**Supervision:** HIROSHI HIRATA, Nakamori Fujii, Kosuke Shimizu, Keita Kobayashi, Takahide Hayano, Yoshiyuki Asai, Koji Shiraishi.

**Validation:** Chieko Baba, HIROSHI HIRATA, Kosuke Shimizu, Keita Kobayashi, Yoshiyuki Asai.

**Visualization:** Koichiro Hiyoshi.

**Writing – original draft:** Chieko Baba.

**Writing – review & editing:** HIROSHI HIRATA, Koji Shiraishi.

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
