## [Decision Letter · Decision Letter 0]

1 Dec 2025

Dear Dr. HIRATA,

Thank you for submitting your manuscript to PLOS ONE. After careful consideration, we feel that it has merit but does not fully meet PLOS ONE’s publication criteria as it currently stands. Therefore, we invite you to submit a revised version of the manuscript that addresses the points raised during the review process.

We look forward to receiving your revised manuscript.

Kind regards,

Mahbub Hasan, PhD

Academic Editor

PLOS ONE

Journal Requirements:

Reviewers' comments:

Reviewer's Responses to Questions

**Comments to the Author**

1. Is the manuscript technically sound, and do the data support the conclusions?

Reviewer #1: Partly

Reviewer #2: Yes

Reviewer #3: Yes

2. Has the statistical analysis been performed appropriately and rigorously?

Reviewer #1: Yes

Reviewer #2: Yes

Reviewer #3: Yes

3. Have the authors made all data underlying the findings in their manuscript fully available?

Reviewer #1: No

Reviewer #2: Yes

Reviewer #3: Yes

4. Is the manuscript presented in an intelligible fashion and written in standard English?

Reviewer #1: Yes

Reviewer #2: Yes

Reviewer #3: Yes

Reviewer #1: This paper investigates the potential role of PSMB8-AS1, a long non-coding RNA (lncRNA), as a biomarker for renal cell carcinoma (RCC). While the study presents an intriguing hypothesis, the current data are preliminary and do not sufficiently support the authors’ conclusions. The proposed mechanism remains unclear. Additionally, the figures and their legends require significant revision, as many are missing essential descriptions and key information.

Specific comments:

1) It is not clear why PSMB8-AS1 was selected among the 1342 upregulated lncRNAs (Fig 1A). Including a table that displays the fold-change in lncRNA expression across the four sample types (ICI responder-tumor/normal and ICI-non responder-tumor/normal) would clarify the rationale for its selection. Please include which data point represents PSMB8-AS1 in the figure 1A.

2) The Y axis should display relative expression (fold) rather than 2-deltaCt values. Given the high variability in expression among ccRCC samples, presenting matched comparisons between tumor and normal tissues from the same patients would be more informative than relying solely on aggregated data.

3) In Figs 2-4, these figures lack sufficient experimental details and explanation of acronyms. The legends should clearly define terms such as pT1-pT4, pN0-pN2, pM0-1, H, and L to ensure clarity for readers.

4) The manuscript would be strengthened by showing that PSMB8-AS1 knockdown in normal renal cells (RPTEC) doesn’t affect cell proliferation, thereby demonstrating that its silencing does not induce general toxicity (Fig 5).

5) If PSMB8-AS1 functions via miR-204/211 and TFAP2A, the study should be assessed whether miR-204/211 are indeed downregulated, and TFAP2A is upregulated in at least some of the normal and tumor tissues from ccRCC patients (Fig 1B).

Minor points:

1) There is a typo in the legend of Fig 3: © should be replaced with (C).

Reviewer #2: The manuscript by Baba et al. presents a well-conducted investigation into the clinical and molecular significance of lncRNA PSMB8-AS1 in clear cell renal cell carcinoma (ccRCC). The work provides novel insights into the PSMB8-AS1/miR-204/211–TFAP2A regulatory axis and its potential as a prognostic biomarker for postoperative recurrence.

However, several aspects could be strengthened to improve the robustness of the conclusions and enhance the translational value of the study.

Comment 1. The study convincingly shows that PSMB8-AS1 regulates the miR-204/211–TFAP2A axis in vitro; however, no functional rescue assays were performed. To firmly establish causality, rescue experiments (e.g., PSMB8-AS1 knockdown combined with TFAP2A overexpression) are recommended to confirm whether the observed phenotypes are TFAP2A-dependent.

While the authors propose that PSMB8-AS1 may modulate immune checkpoint resistance via TFAP2A and PD-L1, direct experimental evidence for PD-L1 regulation is lacking. The inclusion of PD-L1 expression assays (RT-qPCR, western blot, or reporter assays) following PSMB8-AS1 modulation would substantiate this mechanistic link.

C2. The authors focus on miR-204/211 as downstream targets; however, PSMB8-AS1 may regulate additional miRNAs or pathways. Performing transcriptome-wide (RNA-seq) or proteomic profiling after PSMB8-AS1 knockdown could identify other relevant pathways and strengthen the mechanistic depth of the study.

C3. All functional assays were performed in two RCC cell lines (769-P, ACHN). While results are consistent, this limited scope may not capture tumor heterogeneity. Validation in additional RCC models, including patient-derived cells or organoids, would increase generalizability. Furthermore, in vivo studies (xenograft or orthotopic models) are necessary to confirm whether PSMB8-AS1 suppression can reduce tumor growth or metastasis.

C4. The study demonstrates the prognostic value of PSMB8-AS1 using ROC and Cox analyses; however, its clinical applicability remains preliminary. External validation in an independent cohort or cross-validation within the current dataset would increase the reliability of the cutoff thresholds. Integration of PSMB8-AS1 with existing clinical variables (e.g., stage, grade, and treatment) in a nomogram or multivariable prediction model could further support its clinical utility.

In addition, exploring non-invasive detection of PSMB8-AS1 (e.g., in serum or exosomal RNA) would be an important next step toward biomarker translation.

C5. The authors suggest an association between PSMB8-AS1 and immune checkpoint resistance, yet no immune-related assays were performed. Additional analyses such as immune infiltration profiling (TIMER, CIBERSORT) or co-culture experiments with immune cells could clarify how PSMB8-AS1 influences tumor–immune interactions.

C6. RNA was extracted from FFPE samples, which may compromise RNA integrity. It would be beneficial to report RNA quality metrics (e.g., DV200 distribution) for transparency. Validation using fresh-frozen samples or independent qPCR controls could confirm data robustness.

Reviewer #3: The authors identified an as yet under-characterized lncRNA in RCC, and explored its prognostic value and role in oncogenesis. The manuscript is well-written and the findings are novel and well-supported by the experimental designs. This work opens the door for further investigation.

Minor comment: for figures 5 and 6, it would be better to have the graph show individual data points rather than just the mean ± variation. Furthermore, it is unclear if the data is shown as mean±s.d. ,±SEM, or ± some other indicator of variation. This, the n numbers, and other information such as what ** vs **** means, should be indicated in the figure legend and/or elsewhere in the manuscript.

**Do you want your identity to be public for this peer review?** For information about this choice, including consent withdrawal, please see our Privacy Policy

Reviewer #1: No

Reviewer #2: **Yes:** Chul Park

Reviewer #3: No

---

## [Author Response · Author response to Decision Letter 1]

28 Jan 2026

Rebuttal Letter

Dear Editor,

We thank you and the reviewers for your careful evaluation of our manuscript entitled:

“Long non-coding RNA PSMB8-AS1 as a potential biomarker for postoperative recurrence in patients with Fuhrman grades 1–3 clear cell renal cell carcinoma.”

We greatly appreciate the constructive feedback and have revised the manuscript accordingly. Below is our point-by-point response to the reviewer’s and editor’s comments.

Reviewer Comments and Author Responses

Reviewer #1: This paper investigates the potential role of PSMB8-AS1, a long non-coding RNA (lncRNA), as a biomarker for renal cell carcinoma (RCC). While the study presents an intriguing hypothesis, the current data are preliminary and do not sufficiently support the authors’ conclusions. The proposed mechanism remains unclear. Additionally, the figures and their legends require significant revision, as many are missing essential descriptions and key information.

Specific comments:

Comment 1: It is not clear why PSMB8-AS1 was selected among the 1342 upregulated lncRNAs (Fig 1A). Including a table that displays the fold-change in lncRNA expression across the four sample types (ICI responder-tumor/normal and ICI-non responder-tumor/normal) would clarify the rationale for its selection. Please include which data point represents PSMB8-AS1 in the figure 1A.

Response:

We thank the reviewer for this important comment. We agree that the rationale for selecting PSMB8-AS1 among the 1,342 upregulated lncRNAs was not sufficiently clear in the original version.

In the revised manuscript, we have clarified that PSMB8-AS1 was prioritized based on the following criteria: (i) consistent upregulation in tumor tissues compared with matched normal tissues in both ICI responder and non-responder groups, (ii) a relatively large fold-change across the four sample categories, and (iii) limited prior characterization of its biological role in RCC in Result section (line 194-198).

To improve transparency, we have added a supplementary table summarizing fold-change values of representative upregulated lncRNAs across the four groups (ICI responder-tumor/normal and ICI non-responder-tumor/normal), including PSMB8-AS1.

In addition, we have revised Figure 1A to explicitly label the data point corresponding to PSMB8-AS1, allowing readers to readily identify its position within the overall distribution.

Comment 2:

The Y axis should display relative expression (fold) rather than 2-deltaCt values. Given the high variability in expression among ccRCC samples, presenting matched comparisons between tumor and normal tissues from the same patients would be more informative than relying solely on aggregated data.

Response:

We thank the reviewer for this constructive comment. We agree that presentation of relative expression levels improves interpretability. Accordingly, we have revised the y-axis to display relative expression (fold change) instead of 2^−ΔCt values in the revised figure 1B.

In addition, to account for inter-patient variability in ccRCC, we have included paired analyses comparing tumor and matched normal tissues from the same patients. These revisions provide a more informative and biologically relevant representation of the expression differences.

Comment 3:

In Figs 2-4, these figures lack sufficient experimental details and explanation of acronyms. The legends should clearly define terms such as pT1-pT4, pN0-pN2, pM0-1, H, and L to ensure clarity for readers.

Response:

We thank the reviewer for this helpful comment.

To improve clarity, we have revised the figure legends for Figures 2–4 to clearly define all acronyms and terms, including pT stage (pT1–pT4), pathological nodal status (pN0–pN2), metastatic status (pM0–pM1), and the definitions of high (H) and low (L) PSMB8-AS1 expression groups.

Comment 4:

The manuscript would be strengthened by showing that PSMB8-AS1 knockdown in normal renal cells (RPTEC) doesn’t affect cell proliferation, thereby demonstrating that its silencing does not induce general toxicity (Fig 5).

Response:

We appreciate the reviewer’s insightful comment regarding potential general toxicity. We agree that assessment of PSMB8-AS1 knockdown in normal renal epithelial cells would further strengthen the study.

In the present study, our functional analyses were focused on tumor cells to characterize the oncogenic role of PSMB8-AS1 in RCC. Evaluation of PSMB8-AS1 silencing in normal renal epithelial cells was beyond the scope of the current investigation. We have therefore revised the Discussion to acknowledge this limitation and to indicate that assessment of potential effects in normal cells will be an important direction for future studies as follows:

Although our results demonstrate that PSMB8-AS1 promotes malignant phenotypes in RCC cells, the effects of PSMB8-AS1 suppression in normal renal epithelial cells were not evaluated in this study. Future studies examining potential cytotoxic effects in normal cells will be necessary to further assess the specificity and safety of targeting PSMB8-AS1. Notably, PSMB8-AS1 expression was low in normal renal tissues and RPTEC cells, suggesting a potential therapeutic window; however, this hypothesis requires direct functional validation.

すみ

Comment 5:

If PSMB8-AS1 functions via miR-204/211 and TFAP2A, the study should be assessed whether miR-204/211 are indeed downregulated, and TFAP2A is upregulated in at least some of the normal and tumor tissues from ccRCC patients (Fig 1B).

Response:

We thank the reviewer for this important and constructive suggestion.

In response, we analyzed paired tumor and adjacent normal renal tissues from a subset of ccRCC patients to evaluate the expression of PSMB8-AS1, miR-204, and TFAP2A. Quantitative RT-PCR analysis demonstrated that PSMB8-AS1 and TFAP2A were upregulated, whereas miR-204 was downregulated in tumor tissues compared with matched normal tissues. Statistical analysis was performed using the Wilcoxon signed-rank test.

As suggested, we have updated the Statistical Analysis section to include a clear description of the statistical method used for paired analyses, specifically the Wilcoxon signed-rank test.

These results have been added to the Results section and are presented in a revised Figure 1C.

Minor points:

1) There is a typo in the legend of Fig 3: © should be replaced with (C).

Response:

We thank the reviewer for pointing out this typo. We have corrected the legend of Fig. 3 by replacing “©” with “(C)”.

Reviewer 2

Comment 1: The study convincingly shows that PSMB8-AS1 regulates the miR-204/211–TFAP2A axis in vitro; however, no functional rescue assays were performed. To firmly establish causality, rescue experiments (e.g., PSMB8-AS1 knockdown combined with TFAP2A overexpression) are recommended to confirm whether the observed phenotypes are TFAP2A-dependent.

While the authors propose that PSMB8-AS1 may modulate immune checkpoint resistance via TFAP2A and PD-L1, direct experimental evidence for PD-L1 regulation is lacking. The inclusion of PD-L1 expression assays (RT-qPCR, western blot, or reporter assays) following PSMB8-AS1 modulation would substantiate this mechanistic link.

Response:

We thank the reviewer for this important and constructive comment.

In response to the reviewer’s comment, we performed functional rescue experiments demonstrating that forced overexpression of TFAP2A significantly restored the cell proliferation and invasive capacity of RCC cells suppressed by PSMB8-AS1 knockdown (Fig. 7). We have also added a detailed description of the TFAP2A overexpression and rescue experiments to the Methods and Result section to improve clarity and reproducibility.

These results provide direct evidence that the pro-invasive effects of PSMB8-AS1 are mediated, at least in part, through TFAP2A, thereby strengthening the causal relationship within the PSMB8-AS1–miR-204/211–TFAP2A axis.

Regarding PD-L1 regulation, we agree that direct experimental validation would further strengthen the proposed mechanistic link. Although PD-L1 expression assays were not included in the present study, we have revised the Discussion section to clearly state that the involvement of PD-L1 is inferred from previous reports describing TFAP2A-mediated transcriptional regulation of PD-L1 and should therefore be considered a hypothesis requiring future experimental validation.

Discussion Line 281-285

Comment 2: The authors focus on miR-204/211 as downstream targets; however, PSMB8-AS1 may regulate additional miRNAs or pathways. Performing transcriptome-wide (RNA-seq) or proteomic profiling after PSMB8-AS1 knockdown could identify other relevant pathways and strengthen the mechanistic depth of the study.

Response: We agree that PSMB8-AS1 may regulate additional miRNAs or signaling pathways beyond those examined in this study. Transcriptome-wide analyses, such as RNA sequencing, would be valuable to further elucidate the broader regulatory network of PSMB8-AS1. This point has now been discussed as a limitation in the Discussion section.

313-314

Comment 3: All functional assays were performed in two RCC cell lines (769-P, ACHN). While results are consistent, this limited scope may not capture tumor heterogeneity. Validation in additional RCC models, including patient-derived cells or organoids, would increase generalizability. Furthermore, in vivo studies (xenograft or orthotopic models) are necessary to confirm whether PSMB8-AS1 suppression can reduce tumor growth or metastasis.

Response: We agree with the reviewer that the use of only two RCC cell lines represents a limitation of this study. This point has now been acknowledged in the Discussion section, where we also highlight the need for validation in additional RCC models, including patient-derived cells, organoids, and in vivo xenograft or orthotopic models.

(line 315-319)

Comment 4: The study demonstrates the prognostic value of PSMB8-AS1 using ROC and Cox analyses; however, its clinical applicability remains preliminary. External validation in an independent cohort or cross-validation within the current dataset would increase the reliability of the cutoff thresholds. Integration of PSMB8-AS1 with existing clinical variables (e.g., stage, grade, and treatment) in a nomogram or multivariable prediction model could further support its clinical utility.

In addition, exploring non-invasive detection of PSMB8-AS1 (e.g., in serum or exosomal RNA) would be an important next step toward biomarker translation.

Response:

We agree that while ROC and Cox analyses indicate the prognostic potential of PSMB8-AS1, its clinical applicability should be interpreted as preliminary. In the present study, cutoff values were derived in an exploratory manner within a single cohort. We have therefore revised the manuscript to clarify this point and to avoid overinterpretation of the clinical utility.

We also acknowledge that external validation or internal cross-validation would further strengthen the reliability of the cutoff thresholds. These analyses were beyond the scope of the current study and are planned for future investigations.

In addition, although integration of PSMB8-AS1 into a comprehensive nomogram and evaluation of non-invasive detection strategies (e.g., serum or exosomal RNA) would be valuable for clinical translation, such analyses were not feasible in the current dataset. These important directions have now been clearly stated in the revised Discussion.

Line 282-287

Comment 5: The authors suggest an association between PSMB8-AS1 and immune checkpoint resistance, yet no immune-related assays were performed. Additional analyses such as immune infiltration profiling (TIMER, CIBERSORT) or co-culture experiments with immune cells could clarify how PSMB8-AS1 influences tumor–immune interactions.

Response:

We appreciate the reviewer’s insightful comment regarding the potential association between PSMB8-AS1 and immune checkpoint resistance. We agree that immune-related analyses, such as immune infiltration profiling or co-culture experiments, would provide valuable mechanistic insights into tumor–immune interactions.

In the present study, our primary aim was to characterize the tumor-intrinsic role of PSMB8-AS1 and its regulatory axis involving miR-204/211 and TFAP2A. Immune-focused analyses were beyond the scope of the current investigation. We have therefore revised the manuscript to clearly state that the proposed link between PSMB8-AS1 and immune checkpoint resistance remains speculative and requires further validation.

We have also added a discussion highlighting immune infiltration analyses and functional immune assays as important directions for future studies.

312-316

Comment 6: RNA was extracted from FFPE samples, which may compromise RNA integrity. It would be beneficial to report RNA quality metrics (e.g., DV200 distribution) for transparency. Validation using fresh-frozen samples or independent qPCR controls could confirm data robustness.

Response:

We thank the reviewer for this important comment regarding RNA quality in FFPE-derived samples. We acknowledge that RNA extracted from FFPE tissues may exhibit reduced integrity compared with fresh-frozen samples.

In the present study, all FFPE samples were processed using a standardized protocol, and RT-qPCR assays were designed to amplify short amplicons, which is an established approach for mitigating RNA degradation in FFPE-derived RNA. We have revised the manuscript to explicitly acknowledge this limitation and to clarify the methodological considerations taken to ensure data reliability.

Although RNA quality metrics such as DV200 distribution and validation using fresh-frozen samples would further strengthen the robustness of the data, these analyses were beyond the scope of the current study. We have now added this point to the Discussion as an important consideration for future studies. Line 342-348

Reviewer 3

The authors identified an as yet under-characterized lncRNA in RCC, and explored its prognostic value and role in oncogenesis. The manuscript is well-written and the findings are novel and well-supported by the experimental designs. This work opens the door for further investigation.

Minor comment: for figures 5 and 6, it would be better to have the graph show individual data points rather than just the mean ± variation. Furthermore, it is unclear if the data is shown as mean±s.d. ,±SEM, or ± some other indicator of variation. This, the n numbers, and other information such as what ** vs **** means, should be indicated in the figure legend and/or elsewhere in the manuscript.

Response:

We thank the reviewer for this helpful suggestion.

In accordance with the reviewer’s suggestion, we revised the figures to display individual data points along with the mean ± SD. We also clarified in the figure legends that error bars represent standard deviation, specified the sample size (n), and defined the statistical significance levels.

We hope these revisions sufficiently address all concerns, and we thank the reviewers and editorial team again for their valuable input. We believe the revised manuscript has been substantially improved and is now suitable for publication.

Sincerely,

Dr. Hiroshi Hirata, MD, PhD

Department of Urology, Yamaguchi University

Email: hiro1333@yamaguchi-u.ac.jp

---

## [Decision Letter · Decision Letter 1]

15 Feb 2026

Long non-coding RNA PSMB8-AS1 as a potential biomarker for postoperative recurrence in patients with Fuhrman grades 1–3 clear cell renal cell carcinoma

PONE-D-25-32611R1

Dear Dr. HIRATA,

We’re pleased to inform you that your manuscript has been judged scientifically suitable for publication and will be formally accepted for publication once it meets all outstanding technical requirements.

Kind regards,

Mahbub Hasan, PhD

Academic Editor

PLOS One

Additional Editor Comments (optional):

Reviewers' comments:

Reviewer's Responses to Questions

**Comments to the Author**

Reviewer #1: All comments have been addressed

2. Is the manuscript technically sound, and do the data support the conclusions?

Reviewer #1: Yes

3. Has the statistical analysis been performed appropriately and rigorously?

Reviewer #1: Yes

4. Have the authors made all data underlying the findings in their manuscript fully available?

Reviewer #1: Yes

5. Is the manuscript presented in an intelligible fashion and written in standard English?

Reviewer #1: Yes

Reviewer #1: Most concerns have been addressed satisfactorily, and the manuscript is now acceptable for publication.

**Do you want your identity to be public for this peer review?** For information about this choice, including consent withdrawal, please see our Privacy Policy

Reviewer #1: No

---

## [Editor Report · Acceptance letter]

PONE-D-25-32611R1

PLOS One

Dear Dr. HIRATA,

I'm pleased to inform you that your manuscript has been deemed suitable for publication in PLOS One. Congratulations! Your manuscript is now being handed over to our production team.

Kind regards,

on behalf of

Dr. Mahbub Hasan

Academic Editor

PLOS One